# Intestinal Colonization of *Campylobacter jejuni* and Its Hepatic Dissemination Are Associated with Local and Systemic Immune Responses in Broiler Chickens

**DOI:** 10.3390/microorganisms11071677

**Published:** 2023-06-28

**Authors:** Sophie Chagneau, Marie-Lou Gaucher, Philippe Fravalo, William P. Thériault, Alexandre Thibodeau

**Affiliations:** 1Research Chair in Meat Safety, Department of Pathology and Microbiology, Faculté de Médecine Vétérinaire, Université de Montréal, Saint-Hyacinthe, QC J2S 2M2, Canada; 2Swine and Poultry Infectious Diseases Research Center, Department of Pathology and Microbiology, Faculté de Médecine Vétérinaire, Université de Montréal, Saint-Hyacinthe, QC J2S 2M2, Canada; 3Chaire Agroalimentaire du Conservatoire National des Arts et Métiers, 22440 Ploufragan, France; 4Centre de Recherche en Santé Publique (CReSP), Université de Montréal, Montréal, QC H3N 1X9, Canada

**Keywords:** *Campylobacter jejuni*, broiler chickens, intestinal colonization, liver dissemination, cecal tonsils, local immune response, systemic immune response, serum amyloid A, anti-*C. jejuni* IgY antibodies

## Abstract

*Campylobacter jejuni* is an important foodborne pathogen. Despite the lack of clinical signs associated with its colonization in poultry, it has been reported to interact with the intestinal immune system. However, little is known about the interaction between *C. jejuni* and the chicken immune system, especially in the context of hepatic dissemination. Therefore, to follow up on our previous study showing intestinal colonization and hepatic spread of *C. jejuni*, cecal tonsils and liver samples were collected from these birds to determine the mRNA levels of chemokines and cytokines. Serum samples were also collected to determine serum amyloid A (SAA) concentrations and specific IgY titers. Lack of Th17 induction was observed in the cecal tonsils of only the liver-contaminated groups. This hepatic dissemination was accompanied by innate, Th1 and Th2 immune responses in livers, as well as an increase in SAA concentrations and specific IgY levels in sera. *Campylobacter* appears to be able to restrain the induction of the chicken gut immunity in particular conditions, possibly enhancing its hepatic dissemination and thus eliciting systemic immune responses. Although *Campylobacter* is often recognized as a commensal-like bacterium in chickens, it seems to modulate the gut immune system and induce systemic immunity.

## 1. Introduction

*Campylobacter jejuni* causes gastroenteritis in humans and can lead to chronic sequela such as Guillain–Barré and Miller–Fisher syndromes [1]. The worldwide incidence of *C. jejuni* has been increasing in recent years, making it one of the leading causes of bacterial foodborne illnesses [2,3,4]. For example, in the United States, the incidence rate increased by 13% in 2019 compared to the three previous years (2016–2018) [5]. Campylobacteriosis is primarily caused directly or indirectly by the consumption of undercooked or improperly handled poultry products [6]. Although most *C. jejuni* infections are sporadic, outbreaks have been reported after consumption of undercooked chicken livers [7,8].

In contrast to humans, chickens are known to carry high *C. jejuni* loads in their ceca (up to 10^9^ CFU/g of cecal content), largely without showing any clinical signs. However, it was demonstrated that *C. jejuni* can stimulate the intestinal immune system of chickens through its recognition by Toll-like receptors (TLRs) [9]. TLRs are located on the cell surface or on the endolysosomal membrane of cells lining the chicken intestinal epithelium [10]. *C. jejuni* lipooligosaccharide (LOS), lipopeptides, and unmethylated cytosine–phosphate–guanine nucleotide DNA motifs are recognized by the chicken’s TLR4, TLR2, and TLR21, respectively [9]. This recognition induces an innate immune response in broiler chickens characterized by the increased expression of various pro-inflammatory chemokine and cytokine-encoding genes in the ileal and cecal tissues (*CXCLi2*, *IL-6*, and *IL-1β*) [11,12,13,14,15,16], by a modulation of the host defense peptide gene expression in the intestinal tissues [17,18,19], and by an increase in the expression of the *inducible nitric oxide synthase* (*iNOS*)-encoding gene in cecal tissue [20]. Furthermore, this early immune response leads to the infiltration of heterophils in ileal and cecal tissues [11]. Following this innate immune response in *Campylobacter*-colonized birds, the Th17 pathway is predominantly activated in response to CXCLi2, IL-6, and IL-1β production [13,14]. Th17 cells are abundant in the intestinal lamina propria; they prevent the entry of pathogenic bacteria and maintain the integrity of the intestinal barrier by secreting pro-inflammatory cytokines, including IL-17A, IL-17F, and IL-22 [21]. An anti-inflammatory response has also been observed in avian cecal tissues, induced by the expression of *IL-10* [12,13,16,22], a key regulatory cytokine designed to restrain inflammation and ensure immune homeostasis. Lack of *IL-10* expression may cause mucosal damage and diarrhea in *Campylobacter*-colonized chickens [12]. However, the local immune response appears to depend on the chicken genetic line [15,23,24], diet [15], and intestinal microbiota composition [25], on the *C. jejuni* strain [24,26], and on the timing of *C. jejuni* exposure [14,15,26].

Studies have shown that *Campylobacter* can induce systemic immunity after colonization of poultry through an increase in the number of circulating monocytes/macrophages in the chicken peripheral blood [27], in the anti-*C. jejuni* IgY levels in chicken sera [22,24,26,28], and in the α-1-acid glycoprotein (an acute phase protein) concentration in turkey sera [29]. In addition, intra-abdominal injection of *C. jejuni* in Leghorn chickens caused a rapid rise in the number of heterophils infiltrating the abdominal cavity [30]. Jennings et al. [31] observed an activation of T cell responses in the livers of *Campylobacter*-inoculated birds. Interestingly, some authors observed that *C. jejuni* translocation was accompanied by an absence or a downregulation of the cecal immunity and the induction of an innate immune response in the spleens of birds [20,22,32]. 

Although multiple *C. jejuni* genotypes are simultaneously present in the cecal content of most infected broilers [33,34,35], previous immunological studies have been conducted using a single *C. jejuni* strain. Therefore, research focusing on the local immune response in the context of a co-colonization by different *C. jejuni* strains is needed. Moreover, although recent scientific evidence has demonstrated the pathogen’s ability to cross the intestinal barrier and to invade internal organs in broilers [20,24,36,37,38], the link with local and systemic immune responses remains unclear. To address this absence in the literature, we studied the local and systemic immune responses in chickens associated with intestinal colonization by one or two *C. jejuni* strains as well as the hepatic dissemination of *C. jejuni*. We assessed local immunity by analyzing the expression of chemokines, cytokines, and host defense peptides in cecal tonsils, and we examined systemic immunity by measuring the expression of cytokines in liver, the production of the major acute phase protein (serum amyloid A), and the anti-*C. jejuni* IgY antibodies in chicken sera.

## 2. Materials and Methods

### 2.1. Experimental Design and Sample Collection

The current study was a continuation of a previous study designed to investigate the impact of an inoculation of two *C. jejuni* strains on the intestinal colonization and hepatic spread of the bacterium in broiler chickens (certificate issued by the Comité d’Éthique sur l’Utilisation des Animaux: 19-Rech-2039) where details and justification of the experimental designs, as well as detailed results, can be found [37]. Briefly, the two *Campylobacter jejuni* strains used in this study—identified as G2008b and D2008b—were previously isolated from the ceca of commercial broiler chickens and characterized in our laboratory as a strong and a weak competitor for the colonization of the chicken gut, respectively [39]. A total of 197 one-day-old chickens were divided into 8 groups that were placed in two different experimental rooms at the Centre de Recherche Avicole of the Faculté de Médecine Vétérinaire of the Université de Montréal. In both rooms, each group was housed in individual pens that were separated by plexiglass and animals were raised on wood shavings. Birds had ad libitum access to feed (standard mash commercial formulation) and water. They were orally inoculated at 14 days old. Four groups were housed in room 1: (1) not inoculated (control #1); (2) inoculated with 10^3^ CFU of D2008b (10^3^ D2008b); (3) inoculated with 10^3^ CFU of G2008b (10^3^ G2008b); and (4) inoculated with 10^3^ CFU of both strains at the same time (mix #1). Four other groups were housed in room 2: (5) not inoculated (control #2); (6) inoculated with 10^7^ CFU of D2008b (10^7^ D2008b); (7) inoculated with 10^3^ CFU of G2008b (10^3^ G2008b); and (8) inoculated with 10^7^ CFU of D2008b and 10^3^ CFU of G2008b at the same time (mix #2). Necropsies were performed on 8 or 9 birds per group at 1 dpi (days post inoculation), 7 dpi, and 21 dpi. Before cervical dislocation, 5 mL of blood was collected by cardiac puncture at 7 dpi and 21 dpi. Blood samples were incubated 2 h at room temperature and centrifuged at 1000× *g* for 15 min, and chicken sera were stored at −20 °C. At time of necropsy, about 40 mg of cecal tonsils and liver samples of birds were freshly collected and stored at −80 °C in 1 mL of RNAlater Stabilization Solution (Invitrogen, Burlington, ON, Canada). 

In order to monitor specific IgY levels in chicken sera, hyperimmune sera were required to standardize titer determinations between ELISA plates. Therefore, four additional broiler chickens were immunized intramuscularly at 14, 21, and 28 days old with 10^9^ formalin-killed whole *C. jejuni* (bacterins) suspended in 200 µL of PBS (Oxoid, Ottawa, ON, Canada) containing 300 mg of Quil-A^®^. This experimentation was approved by the Comité d’Éthique sur l’Utilisation des Animaux of the Faculté de Médecine Vétérinaire of the Université de Montréal (certificate number: 20-Rech-2070). Bacterins were obtained by incubating 10^9^ CFU of an equal mix of D2008b and G2008b in formalin (Fisher Scientific, Ottawa, ON, Canada) for 24 h at 4 °C. After five washings, bacterins were plated on tryptic soy blood agar (Fisher Scientific) to verify the absence of bacterial growth and stored at −80 °C until injection. As previously described, 5 mL of blood were collected from birds at 35 days of age using a cardiac puncture before cervical dislocation. Hyperimmunized chicken sera were pooled together and stored in aliquots at −20 °C.

### 2.2. RNA Extraction from Cecal Tonsils and Liver Samples, and Reverse Transcription

RNA extractions were performed as previously described [37], with a mechanic lysis step consisting of two runs of 20 s at 4 m/s for cecal tonsils and of one run of 10 s at 4 m/s for liver samples. RNA was quantified and used for the reverse transcription to cDNA as previously described [37].

### 2.3. Real-Time Quantitative PCR (qPCR) from cDNA

A total of 1 μL of cDNA was amplified in duplicate for each gene and each bird, as previously described [37]. The PCR program consisted of an initial denaturation step of 10 min at 95 °C, followed by 40 cycles of 10 s at 95 °C, 10 s at an annealing temperature indicated in Table 1, 10 s at 72 °C, and a final step of high-resolution melting to verify the specificity of the PCR products. The relative expression levels of the target genes were evaluated as previously described [37].

### 2.4. Quantification of Serum Amyloid A in Sera

Serum amyloid A (SAA) concentration in chicken sera was measured using a commercial ELISA kit: chicken serum amyloid A ELISA (Life diagnostics Inc., West Chester, PA, USA). According to the manufacturer’s instructions, ELISA was performed in duplicate for each standard and diluted serum sample. Absorbances were read at 450 nm using a Biochrom EZ Read 400 Microplate Reader (Biochrom, Cambridge, UK), and the concentrations were determined from the standard curve by a four-parameter logistic regression.

### 2.5. Determination of Anti-C. jejuni IgY Titers in Sera

The *C. jejuni* strains G2008b and D2008b were grown on tryptic soy blood agar plates (Fisher Scientific) for 24 h at 42 °C under microaerobic conditions using the gas pack CampyGen system (Oxoid) and suspended in PBS (Oxoid). Total *C. jejuni* proteins were obtained by sonication using a Sonics Vibra-Cell VC130 Ultrasonic Processor (Sonics and Materials Inc., Newtown, CT, USA), consisting of five runs of 30 s at 0.4 watts with incubation for 60 s on ice between each run. Cellular debris were removed by centrifugation at 12,000× *g* for 15 min at 4 °C. Proteins were quantified using the Pierce BCA Protein Assay Kit (ThermoFisher Scientific), aliquoted, and stored at −80 °C. The ELISA plates were coated overnight at 4 °C with 100 μL per well of an equal total protein mix from both *C. jejuni* strains at a concentration of 20 μg/mL. Plates were washed five times with PBS containing 0.05% Tween-20 (Sigma-Aldrich, Oakville, ON, Canada) and blocked with 200 μL of 2% BSA (Sigma-Aldrich) for 30 min at room temperature. To determine the *C. jejuni* IgY titers, chicken sera were serially diluted 2-fold in PBS containing 0.05% Tween-20 and 2% BSA, and 100 μL was incubated in plates for 1 h at room temperature. After five washes, 100 μL of the HRP-conjugated goat anti-chicken IgG (IgY) Fc fragment (diluted 1:40,000) (Bethyl Laboratories #A30-104P, Montgomery, TX, USA) was added in wells for 1 h at room temperature. Results were revealed by adding 100 μL of TMB-ELISA Substrate Solution (ThermoFisher Scientific). To avoid variations between plates, the reaction was stopped by adding 100 μL of 0.5 M H_2_SO_4_ when the absorbance of 1.0 at 450 nm was obtained for pooled hyperimmunized chicken sera incubated in each ELISA plate. Absorbances were read using a Biochrom EZ Read 400 Microplate Reader (Biochrom). *C. jejuni* IgY titer of samples was the last dilution of serum with an absorbance ≤0.16 (cut-off).

### 2.6. Statistical Analyses

Figures and statistical analyses were performed with GraphPad Prism 9.5.0 (GraphPad Software Inc., La Jolla, CA, USA). A Shapiro–Wilk test was used to verify data normality. An ANOVA test followed by Tukey’s multiple comparison tests were used to analyze statistical differences of RT-qPCR results between groups within the same time point. A Kruskal–Wallis test followed by Dunn’s post hoc tests were used to analyze statistical differences of ELISA results between chicken groups within the same time point.

## 3. Results

### 3.1. Transcript Levels of Chemokine, Cytokines, and Host Defense Peptides in Cecal Tonsils of Chickens Inoculated with C. jejuni

To investigate the induction of a local immune response in birds after inoculation with *C. jejuni*, as well as its possible association with hepatic spread, we used RT-qPCR to quantify the transcript levels of the *CXCLi2* chemokine and *IL-1β*, *IL-17A* cytokines, which are all markers of the Th17 immune response in cecal tonsils at the onset of the gut colonization (i.e., 1 and 7 dpi) (see Figure 1). In addition, the mRNA amount of the anti-inflammatory cytokine *IL-10* was also quantified in the same samples. At 1 dpi in room 2, only the *CXCLi2* transcript levels were significantly upregulated (*p* < 0.05) in the cecal tonsils from birds inoculated with 10^7^ CFU of D2008b compared to birds from the control group and birds inoculated with the mix of the two *C. jejuni* strains, though its expression decreased at 7 dpi. In room 1, we noted that the mRNA levels of Th17 indicators were only significantly increased in cecal tonsils from chickens inoculated with 10^3^ CFU of G2008b compared to birds from the control group at 7 dpi. The same pattern was observed in birds inoculated with 10^3^ CFU of G2008b and housed in room 2. A significant increase in *IL-10* expression was also noted at 7 dpi in the cecal tonsils of all *C. jejuni*-colonized groups, as opposed to the uncolonized ones.

We also examined the mRNA levels of the host defense peptides *AvBD1* and *CATH2* (see Figure 2) in the same samples. No significant differences in *AvBD1* and *CATH2* transcript levels were detected in cecal tonsils between groups in either room at 1 and 7 dpi. 

### 3.2. Transcript Levels of Chemokine and Cytokines in the Liver of Broiler Chickens Inoculated with C. jejuni

To study the induction of a systemic immune response following the hepatic spread of *C. jejuni* that occurred at 7 dpi and 21 dpi in broiler chickens, we used RT-qPCR to quantify the transcript levels of the pro-inflammatory chemokine *CXCLi2*, *IL-1β* cytokine, and the anti-inflammatory cytokine *IL-10* in the liver samples of birds (see Figure 3). In room 1, we noted a significant upregulation of the *IL-1β* and *IL-10* mRNA levels at 21 dpi in the liver of birds inoculated with the mix, compared to other groups. In room 2, the mRNA amounts of *CXCLi2*, *IL-1β*, and *IL-10* increased in the liver of birds inoculated with the *C. jejuni* mix at 7 dpi compared to birds from the control group and birds inoculated with 10^3^ CFU of G2008b. These amounts decreased at 21 dpi. In this same room, we observed a significant upregulation of *IL-1β* expression at 7 dpi in birds inoculated with 10^7^ CFU of D2008b (*p* < 0.01).

We also examined the mRNA levels of *IFNγ* (as a marker of the Th1 immune response activation) and *IL-13* (as a marker of the Th2 immune response activation) in the same samples (see Figure 4). In room 1, we observed a significant increase at 21 dpi in the levels of these mRNAs in the liver of birds inoculated with the *C. jejuni* mix compared to the other conditions. In room 2, we found that transcript levels of these two targets were significantly upregulated in livers from birds inoculated with the *C. jejuni* mix at 7 dpi compared to others, but levels were decreased at 21 dpi. Furthermore, in the same room, livers from chickens inoculated with 10^7^ CFU of D2008b showed higher mRNA levels of *IFNγ* and *IL-13* at 21 dpi than livers from uninoculated birds and from birds inoculated with 10^3^ CFU of G2008b.

### 3.3. Serum Amyloid A Quantification in Sera of Chickens Inoculated by C. jejuni

Acute phase proteins (APPs) are present in the bloodstream. They are largely synthesized by hepatocytes in response to pro-inflammatory cytokines and are released after infection, inflammation, trauma, neoplasia, or stress [44]. Serum amyloid A (SAA) is the major and most sensitive APP in chickens. Therefore, for the first time, the concentration of this APP was quantified in the sera of chickens inoculated by *C. jejuni* at 1 dpi, 7 dpi, and 21 dpi using a commercial ELISA kit (see Figure 5). No significant differences were observed between the chicken groups in room 1 at 1 dpi, 7 dpi, and 21 dpi. In contrast, in room 2, we found that chickens inoculated with 10^7^ CFU of D2008b at 7 dpi had significantly higher SAA concentrations in serum than those uninoculated (*p* < 0.05) and those inoculated with 10^3^ CFU of G2008b (*p* < 0.05). Interestingly, we observed an increase in SAA concentrations in some birds inoculated with the mix at 7 dpi—similar to the chickens inoculated with 10^7^ CFU of D2008b—despite the absence of significant differences with uninoculated birds. 

### 3.4. Anti-C. jejuni IgY Antibody Levels in Sera of Chickens Inoculated with C. jejuni

To assess the adaptative systemic immune response in broiler chickens associated with the intestinal colonization and hepatic spread of *C. jejuni*, anti-*C. jejuni* IgY titers were measured by ELISA in the sera of birds at 7 and 21 dpi (see Figure 6). Serum samples from birds inoculated with 10^3^ CFU of G2008b at 21 dpi and housed in room 1 revealed IgY levels 9.4-fold higher than titers measured from the control serum. We also noted a 31.6-fold increase in anti-*C. jejuni* IgY antibodies in the serum of birds inoculated with the *C. jejuni* mix compared to titers from the control group (*p* < 0.001). At the same time, in room 2, a similar pattern was observed for birds inoculated with 10^3^ G2008b. Moreover, these IgY levels were significantly higher in birds inoculated with 10^7^ CFU of D2008b (median: 21,516) than in uninoculated birds (median: 1312; *p* < 0.0001) and in birds inoculated with 10^3^ CFU of G2008b (median: 5121; *p* < 0.05). We also observed a significant 13.4-fold increase in IgY levels against *C. jejuni* in chickens inoculated with the *C. jejuni* mix compared to these same levels in uninoculated birds (*p* < 0.001)

To facilitate the comparison and the understanding of all these findings, the main results of our previous study [37] and of the current study were summarized in Table 2 and Table 3.

## 4. Discussion

Due to the lack of effective methods to prevent or reduce the colonization of broiler chickens by *C. jejuni*, its persistence in commercial flocks has become a major public health concern. Nevertheless, little is known about the mechanisms underlying the interaction of *C. jejuni* with the chicken gut immune system, especially in a context of multi-strain colonization, which appears to be common on farms. Although *C. jejuni* appears to be able to cross the intestinal barrier and to invade internal organs in broilers through unknown mechanisms, the relationship between liver invasion and local and systemic immune responses remains unclear. These processes can therefore be better understood by examining the impacts of intestinal colonization by a single *C. jejuni* strain or mixed strains and the impacts of their hepatic spread on the local and systemic immune responses in broilers.

In the previous study conducted by our group [37], no clinical signs were observed and no significant difference in body weight was noted at 1, 7, and 21 dpi. Regarding the gut establishment, we observed that very few birds were colonized by *C. jejuni* at 1 dpi. At 7 dpi and 21 dpi, *C. jejuni* was detected neither in the guts of uninoculated birds housed in both rooms nor in birds inoculated with 10^3^ CFU of D2008b housed in room 1 compared to other chicken groups. As a result of our strain-specific qPCR approach, we confirmed that throughout these animal experiments, birds inoculated with a single *C. jejuni* strain were colonized by that strain only. In room 1 in the caecum of birds inoculated with the mix, we saw D2008b colonization starting from 7 dpi only in the presence of G2008b, and their colonization levels became almost equal at 21 dpi. Therefore, we hypothesized that D2008b benefited from G2008b’s intestinal colonization, which may be related to a commensalism mechanism between *C. jejuni* strains. In room 2, we observed inter-strain competition for gut colonization, characterized by a predominance of G2008b in the ileum and cecum at 21 dpi. In addition, we saw the D2008b translocation to the liver for birds inoculated with the mix and housed in room 1 at 21 dpi in those inoculated with the mix and housed in room 2 at 7 dpi and 21 dpi and in those inoculated with 10^7^ CFU of D2008b and housed in room 2 at 21 dpi without clinical signs throughout the animal experiments. Therefore, we speculated that a high inoculum of *C. jejuni* and/or the presence of G2008b appeared to facilitate and accelerate the hepatic spread of D2008b. Moreover, these hepatic translocations appeared to be accompanied by a moderate and transient decrease in tight junction protein expression levels in the ileum. We therefore concluded that G2008b is more suitable for gut colonization, but it contributed to the establishment of D2008b in the cecum and spread to the liver.

In order to explore the local immune response of broiler chickens colonized with single or mixed strains of *C. jejuni*, we analyzed the expression levels of Th17 pathway indicators (*CXCLi2*, *IL-1β,* and *IL-17A*) and the anti-inflammatory cytokine *IL-10* in the cecal tonsils. At 1 dpi, the *CXCLi2* mRNA levels were only significantly increased in the cecal tonsils of chickens inoculated with 10^7^ CFU of single D2008b, which is consistent with the literature [11,15,20], suggesting that heterophils and macrophages might be attracted to infection sites [45]. Therefore, we hypothesize that with a high bacterial inoculum mixed and distinct strains of *C. jejuni* might have different effects on the timing of the innate immune system activation. The pro-inflammatory cytokine IL-1β could be secreted after CXCLi2 production and would mediate the Th17 signaling pathway with the production of IL-17A in the challenged birds. Interestingly, in the present study, no increase in mRNA levels of Th17 pathway indicators was observed at 7 dpi in the cecal tonsils of liver-contaminated birds. Similar results showing the absence or reduction of a proinflammatory response in cecal tonsils of inoculated birds have been reported [20,22], especially the Th17 response [32] before and/or during the *C. jejuni* translocation in chickens. The absence of Th17 signaling at 7 dpi might compromise the integrity of the intestinal epithelial barrier, which could, in turn, facilitate the reduced expression of tight junction proteins in the presence of *C. jejuni* (as we previously reported [37]) and the ensuing hepatic spread. Furthermore, the significant increase in *IL-10* expression in the cecal tonsils of *C. jejuni*-inoculated birds is in accordance with the literature [14,16,22] and would result in the long-term colonization of *C. jejuni* in the intestine and the lack of observed pathologies [12].

In response to the activation of TLRs, phagocytes and epithelial cells lining the gastrointestinal tract also secrete two major families of host defense peptides (HDPs) in chickens: β-defensins and cathelicidins [46]. Due to their antimicrobial and immunomodulatory properties, HDPs play an important role in the first line of defense against the crossing of the intestinal barrier by pathogens. In the current study, we found that *C. jejuni* did not affect *AvBD1* expression in cecal tonsils at 1 dpi and 7 dpi, except for some birds showing an upregulation. Our findings appear to be consistent with those reported by Li et al. [19], showing no significant differences of *AvBD1* expression at 1, 3, and 15 dpi for chickens inoculated with 10^4^ CFU of *C. jejuni*. Our findings also showed that *CATH2* expression levels were not altered by *C. jejuni* in cecal tonsils of birds at 1 dpi and 7 dpi, which aligns with findings reported by Li et al. [19]. Nonetheless, they contradict those reported by van Dijk et al. [17], who showed a downregulation of *CATH2* expression at 2 dpi in the jejunum of chickens colonized by *C. jejuni*. We hypothesized that *CATH2* expression might be differently altered by *Campylobacter* between the jejunum and cecum probably due to different *C. jejuni* loads and/or microbiota composition, as observed for the expression of several *β-defensins* throughout the gastrointestinal tract of broiler chickens [18]. Furthermore, since *CATH2* is expressed only by heterophils [47], differences in heterophil infiltration between the jejunum and the cecum in response to *C. jejuni* colonization could also explain the observed divergences, as reported by Smith et al. [11]. Given that regulation of *HDPs* expression by *C. jejuni* appears to be tissue-dependent, inoculum-dependent, timepoint-dependent, and possibly strain-dependent, future studies should aim to better characterize the role and control of *HPD* expression in the broiler chicken gut in response to *C. jejuni* infection. 

Analysis of the proinflammatory response in the bird livers revealed an increase in *CXCLi2* and *IL-1β* transcripts for livers contaminated by *C. jejuni* compared with others, which is in keeping with the hepatic dissemination observed in our previous study [37]. However, in the present work, the expression of these markers in the liver of chickens inoculated with 10^7^ CFU of D2008b started to increase at 7 dpi, whereas we only found *C. jejuni* by bacterial culture at 21 dpi. Therefore, we believe that in this group, the translocation would start at 7 dpi with a *C. jejuni* load below the detection limit of 50 CFU/g in the liver. These findings indicate that *C. jejuni* translocation might induce innate immunity in the liver, possibly through its recognition by TLRs-expressing hepatocytes and/or hepatic immune cells. As reported by Vaezirad et al. [20] and Mortada et al. [22], in *C. jejuni*-contaminated chicken spleens after inoculation, the inflammation observed in the present work was also accompanied by an anti-inflammatory response. Consistent with the absence of liver pathology observed in our previous study [37], these results suggest that *C. jejuni* might contain the inflammatory response for its survival and establishment by mediating the induction of regulatory cytokines such as IL-10. However, in our previous study, we noted that the number of *C. jejuni*-contaminated livers seemed to decrease between 7 dpi and 21 dpi. This observation could be attributed to the latter triggering the adaptive immunity to clear intracellular and/or extracellular *C. jejuni* by eliciting cellular Th1 and humoral Th2 immune responses, as indicated by the upregulation of *IFNγ* and *IL-13* expressions in liver-contaminated groups. These results align with the increased proliferation of hepatic T cells observed in livers of *C. jejuni*-inoculated chickens reported by Jennings et al. [31], despite low levels of translocation. Nevertheless, all of these findings remain to be validated by performing additional *in vivo* assays to better understand the role of immunity in *C. jejuni* establishment in the early stages of liver infection and in its late clearance.

APPs (acute phase proteins) are mainly synthesized by hepatocytes in response to proinflammatory cytokines (IL-6, TNF-α and IL-1β) and released into the bloodstream [44]. They play an essential role in the innate immune response to restore homeostasis and to contain microbial infections [48]. Therefore, they are widely used as physiological biomarkers for bacterial and viral infections and as a way to monitor vaccine responses. In chickens, SAA is the major and the most sensitive APP that opsonizes Gram-negative bacteria [49] and induces protective responses from immune cells [50]. In the current study, increased SAA levels were only observed in room 2 at 7 dpi for some birds inoculated with D2008b in the presence or absence of G2008b, corresponding to liver-contaminated groups in this room. According to some authors, SAA concentrations in broiler chicken sera increase during the acute phase infection after exposure to a stimulus, such as vaccination [51] or *Escherichia coli* [52] and *Salmonella* Typhimurium infections [53]. However, its concentrations rapidly drop due to its short half-life, which could explain why very few significant differences were observed in this work. Interestingly, the SAA secretion appears to coincide with the upregulation of *IL-1β* expression in the liver after *C. jejuni* translocation in room 2. According to observations in *Listeria monocytogenes*-infected mice [54], we hypothesize that the hepatic anti-inflammatory response following *C. jejuni* dissemination, described above, might induce SAA synthesis to rapidly control the liver infection. Elevated SAA levels might have contributed to the reduction in the number of *C. jejuni*-contaminated livers observed between 7 and 21 dpi during our previous study [37]. Future studies should focus on examining the effect of *C. jejuni* translocation on APP expression, especially SAA, to better understand the underlying host clearance mechanisms against this microorganism in the liver. 

Finally, the effect of *C. jejuni* infection on adaptative systemic immunity was assessed by measuring the IgY levels against *C. jejuni*. The observed increase in anti-*C. jejuni* IgY titers in challenged birds at 21 dpi is consistent with studies reporting that specific IgY levels started to increase between the second and third week post-inoculation [22,24,26,28]. In this work, the intestinal colonization alone resulted in low specific IgY levels in the serum, whereas liver dissemination of *C. jejuni* increased these levels. Due to the low *C. jejuni* dissemination rate reported in previous studies or the absence of monitoring of its intestinal translocation [22,24,26,28], it is difficult to compare our data with the literature. The slight increase in specific IgY levels in liver-uncontaminated chicken groups might be due to the low intestinal immune response following the *C. jejuni* infection as described above. These results lead us to suspect that intestinal colonization alone of *C. jejuni* does not affect the adaptive systemic immunity much, which is consistent with its commensal-like characteristics. Regarding the liver-contaminated groups, we hypothesize that elevated levels of specific IgY might be associated with the induction of hepatic Th2 immunity in response to *C. jejuni* translocation described above. However, we noted a delay between the initiation of the Th2 immune response in livers and the increase in specific IgY levels in sera, possibly due to the transition from IgM to IgY. Future studies are needed to confirm our findings, especially with other strains.

Taken together, we showed that *C. jejuni* interacts with the host immune system and controls the proinflammatory response in avian cecal tonsils, which, in some cases, decreases to near basal levels. The absence of this proinflammatory response could allow *Campylobacter* to compromise gut integrity, thereby promoting its hepatic spread. When the cecum was colonized by any of the strains used in this study, an anti-inflammatory response was observed, which may have led to the tolerance of *C. jejuni* in the chicken gut. Liver invasion induced an innate immune response, including the expression of proinflammatory cytokines, which would lead to SAA secretion in an attempt to clear the microorganism. In addition, the establishment of *C. jejuni* in the liver of infected birds triggered Th1 and Th2 immune responses, causing the production of specific IgY antibodies that could, in turn, have contributed to decreasing the translocation rate and the hepatic clearance of the bacterium. All of these findings highlight a potential link between *C. jejuni* intestinal colonization, its hepatic dissemination, and associated local and systemic immune responses. 

Although *Campylobacter* is often recognized as a commensal-like bacterium in chickens, it appears to be able to modulate the gut immune system and to induce a systemic immunity, possibly after its hepatic dissemination, without causing clinical signs or affecting chicken performances. Albeit all these findings need to be confirmed by other studies, analysis of SAA concentrations and specific IgY levels in sera could be used to identify and eliminate contaminated livers in slaughterhouses. We also showed that immune response to strains restricted to the gut environment differ from strains that spread to the liver. We therefore recommend monitoring extraintestinal translocation for further studies that aim to look at the immune response of birds to *C. jejuni* colonization, especially studies reporting the results of vaccine challenges. 

## Figures and Tables

**Figure 1 microorganisms-11-01677-f001:**
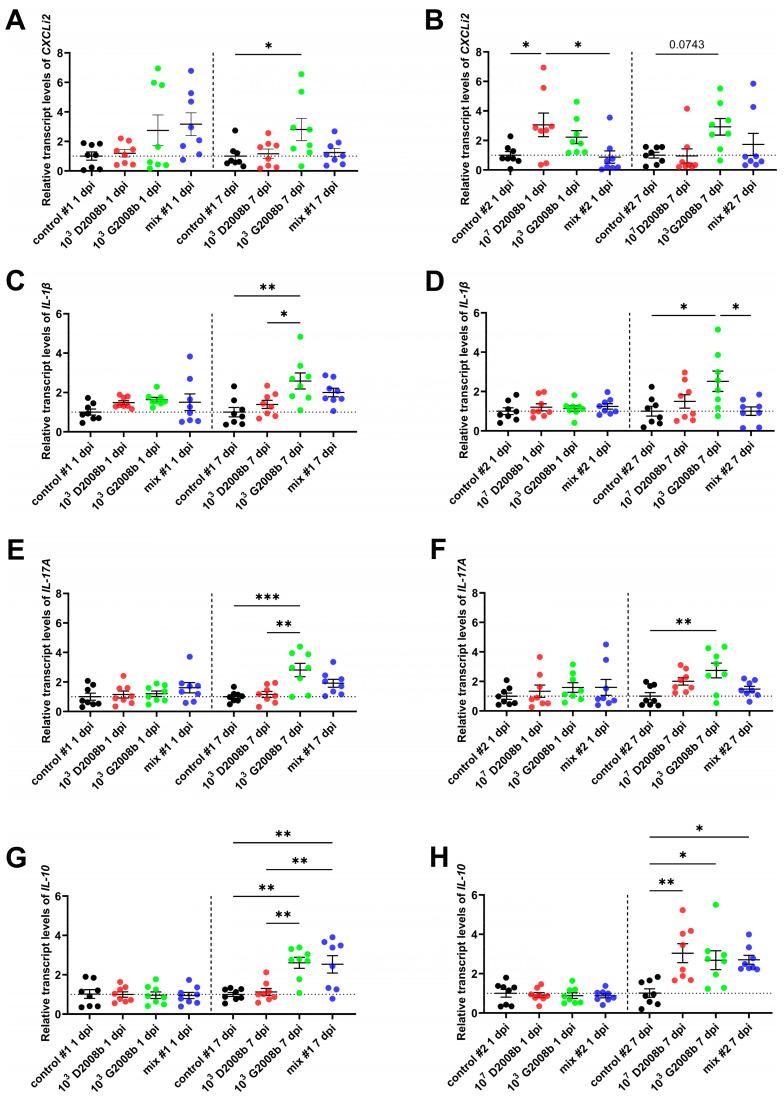
Transcript levels of chemokine *CXCLi2* and cytokines *IL-1β*, *IL-17A*, *IL-10* in cecal tonsils of birds at 1 and 7 dpi: (**A**) relative mRNA amounts of *CXCLi2* in cecal tonsils of birds in room 1; (**B**) relative mRNA amounts of *CXCLi2* in cecal tonsils of birds in room 2; (**C**) relative mRNA amounts of *IL-1β* in cecal tonsils of birds in room 1; (**D**) relative mRNA amounts of *IL-1β* in cecal tonsils of birds in room 2; (**E**) relative mRNA amounts of *IL-17A* in cecal tonsils of birds in room 1; (**F**) relative mRNA amounts of *IL-17A* in cecal tonsils of birds in room 2; (**G**) relative mRNA amounts of *IL-10* in cecal tonsils of birds in room 1; (**H**) relative mRNA amounts of *IL-10* in cecal tonsils of birds in room 2. Data are presented as mean  ±  SEM. *, **, and *** indicate *p* < 0.05, *p* < 0.01, and *p* < 0.001, respectively.

**Figure 2 microorganisms-11-01677-f002:**
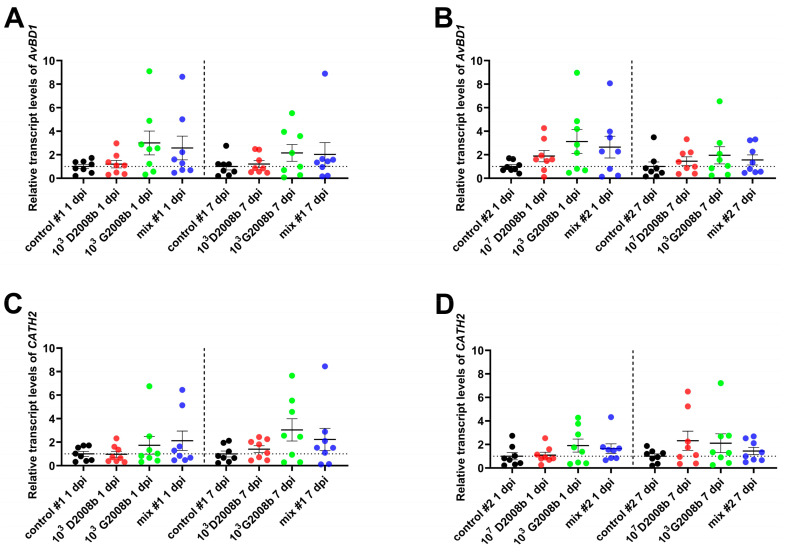
Transcript levels of host defense peptides *AvBD1* and *CATH2* in cecal tonsils of birds at 1 and 7 dpi: (**A**) relative mRNA amounts of *AvBD1* in cecal tonsils of birds in room 1; (**B**) relative mRNA amounts of *AvBD1* in cecal tonsils of birds in room 2; (**C**) relative mRNA amounts of *CATH2* in cecal tonsils of birds in room 1; (**D**) relative mRNA amounts of *CATH2* in cecal tonsils of birds in room 2. Data are presented as mean  ±  SEM.

**Figure 3 microorganisms-11-01677-f003:**
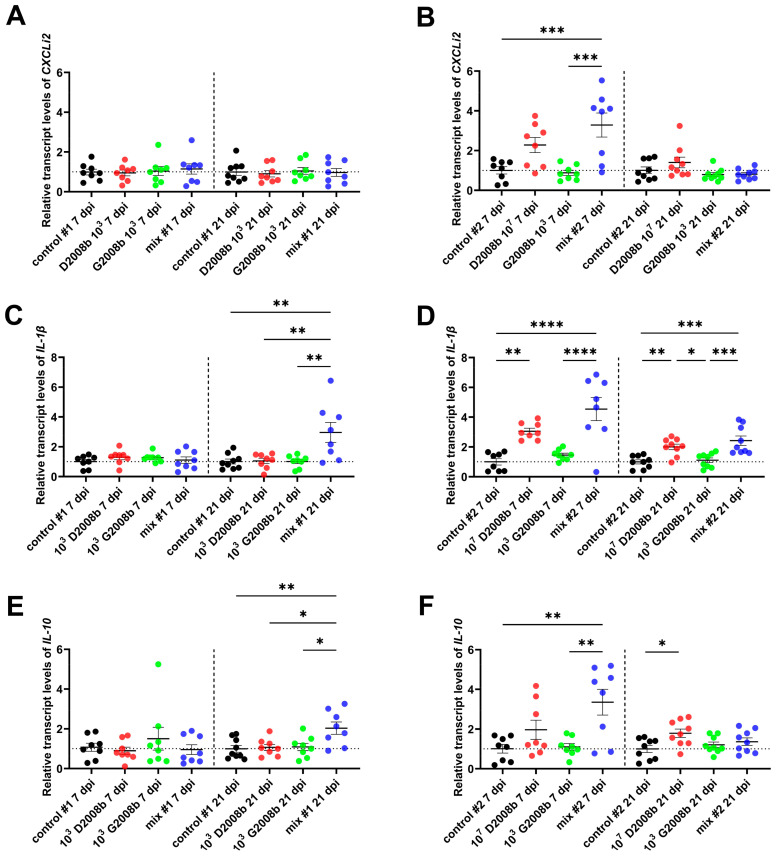
Transcript levels of chemokine *CXCLi2* and cytokines *IL-1β*, *IL-10* in bird liver at 7 and 21 dpi: (**A**) relative mRNA amounts of *CXCLi2* in bird liver in room 1; (**B**) relative mRNA amounts of *CXCLi2* in bird liver in room 2; (**C**) relative mRNA amounts of *IL-1β* in bird liver in room 1; (**D**) relative mRNA amounts of *IL-1β* in bird liver in room 2; (**E**) relative mRNA amounts of *IL-10* in bird liver in room 1; (**F**) relative mRNA amounts of *IL-10* in bird liver in room 2. Data are presented as mean  ±  SEM. *, **, ***, and **** indicate *p* < 0.05, *p* < 0.01, *p* < 0.001, and *p* < 0.0001, respectively.

**Figure 4 microorganisms-11-01677-f004:**
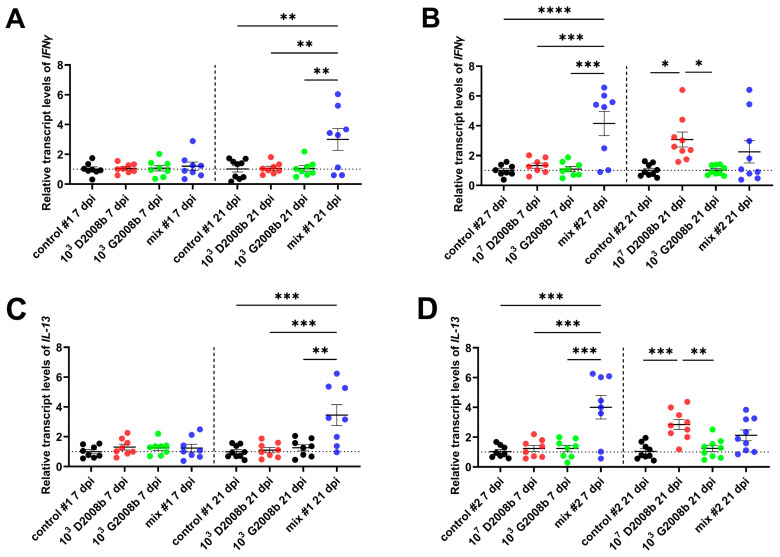
Transcript levels of cytokines *IFNγ* and *IL-13* in bird liver at 7 dpi and 21 dpi: (**A**) relative mRNA amounts of *IFNγ* in bird liver in room 1; (**B**) relative mRNA amounts of *IFNγ* in bird liver in room 2; (**C**) relative mRNA amounts of *IL-13* in bird liver in room 1; (**D**) relative mRNA amounts of *IL13* in bird liver in room 2. Data are presented as mean  ±  SEM. *, **, ***, and **** indicate *p* < 0.05, *p* < 0.01, *p* < 0.001, and *p* < 0.0001, respectively.

**Figure 5 microorganisms-11-01677-f005:**
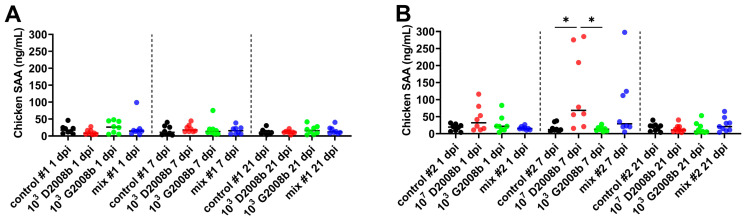
Serum amyloid A concentrations in sera of birds at 1, 7, and 21 dpi: (**A**) serum amyloid A (SAA) concentration in sera of birds in room 1; (**B**) SAA concentration in sera of birds in room 2. Horizontal bars illustrate the median for each group. * indicates *p* < 0.05.

**Figure 6 microorganisms-11-01677-f006:**
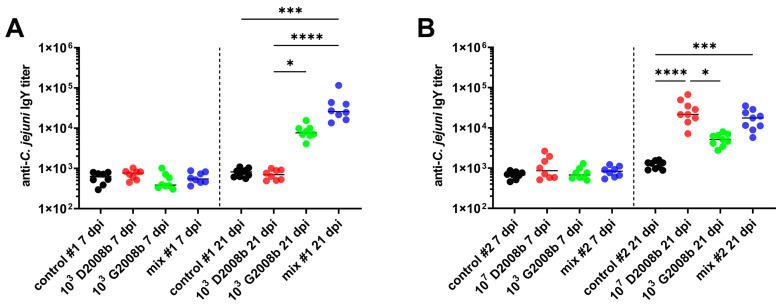
IgY antibody levels against *C. jejuni* in sera of chickens at 7 and 21 dpi: (**A**) anti-*C. jejuni* IgY titers in sera of birds in room 1; (**B**) anti-*C. jejuni* IgY titers in sera of birds in room 2. Horizontal bars illustrate the median for each group. *, ***, and **** indicate *p* < 0.05, *p* < 0.001, and *p* < 0.0001, respectively.

**Table 1 microorganisms-11-01677-t001:** Primer sequences used for RT-qPCR.

Target Gene	Primer Sequences (5′–3′)	Annealing Temperature	Reference
*β-actin*	F: CAACACAGTGCTGTCTGGTGGTA R: ATCGTACTCCTGCTTGCTGATCC	60	[40]
*Ribosomal protein L32 (RPL32)*	F: ATGGGAGCAACAAGAAGACG R: TTGGAAGACACGTTGTGAGC	58	[41]
*Chemokine [C-X-C motif] ligand i2 (CXCLi2)*	F: CCAAGCACACCTCTCTTCCA R: GCAAGGTAGGACGCTGGTAA	60	[40]
*Interleukin-1β (IL-1β)*	F: GTGAGGCTCAACATTGCGCTGTA R: TGTCCAGGCGGTAGAAGATGAAG	63	[40]
*Interleukin-10 (IL-10)*	F: TTTGGCTGCCAGTCTGTGTC R: CTCATCCATCTTCTCGAACGTC	60	[42]
*Interleukin-17A (IL-17A)*	F: CATGGGATTACAGGATCGATGA R: GCGGCACTGGGCATCA	60	[13]
*Interleukin-13 (IL-13)*	F: ACTTGTCCAAGCTGAAGCTGTC R: TCTTGCAGTCGGTCATGTTGTC	60	[40]
*Interferon-γ (IFN-γ)*	F: ACACTGACAAGTCAAAGCCGCACA R: AGTCGTTCATCGGGAGCTTGGC	60	[40]
*Avian β-defensin1 (AvBD1)*	F: GGTTCTTACTGCCTTGCTGT R: TGACTTCCTTCCTAGAGCCT	57	[43]
*Cathelicidin-2 (CATH2)*	F: GATGGTGACCTTAGGGCGGAA R: CGAGATCAATCTACGCTGCAGAG	62	[17]

**Table 2 microorganisms-11-01677-t002:** Summary of the main results of our previous study [37] and of the current study for chickens housed in room 1 at 1, 7, and 21 dpi (unless otherwise specified).

Conditions	Control #1	10^3^ D2008b	10^3^ G2008b	Mix #1
Cecal colonization of *C. jejuni*	No	No	Yes	Yes
Hepatic spread of *C. jejuni*	No	No	No	Yes (at 21 dpi only)
Cecal immunity ^a^		No Th17 induction	Th17 induction at 7 dpi Significant increase in *IL-10* mRNA levels at 7 dpi	No Th17 induction Significant increase in *IL-10* mRNA levels at 7 dpi
Hepatic Immunity ^b^		No induction of immune responses	No induction of immune responses	Significant increase in *IL-1β*, *IL-10*, *IFNγ*, and *IL-13* mRNA levels at 21 dpi
Systemic immunity		No increase in specific IgY levels	No significant increase in specific IgY levels at 21 dpi (compared to control)	Significant increase in specific IgY levels at 21 dpi

^a^ Cecal immunity was assessed only at 1 dpi and 7 dpi ; ^b^ hepatic immunity was assessed only at 7 dpi and 21 dpi.

**Table 3 microorganisms-11-01677-t003:** Summary of the main results of our previous study [37] and of the current study for chickens housed in room 2 at 1, 7, and 21 dpi (unless otherwise specified).

Conditions	Control #2	10^7^ D2008b	10^3^ G2008b	Mix #2
Cecal colonization of *C. jejuni*	No	Yes	Yes	Yes
Hepatic spread of *C. jejuni*	No	Yes (at 21 dpi only)	No	Yes (at 7 and 21 dpi only)
Cecal immunity ^a^		Significant increase in *CXCLi2* mRNA levels at 1 dpi, but no Th17 induction at 7 dpi Significant increase in *IL-10* mRNA levels at 7 dpi	Th17 induction at 7 dpi Significant increase in *IL-10* mRNA levels at 7 dpi	No Th17 induction Significant increase in *IL-10* mRNA levels at 7 dpi
Hepatic Immunity ^b^		Significant increase in *IL-1β* mRNA levels at 7 dpi; and *IL-1β*, *IL-10*, *IFNγ*, and *IL-13* at 21 dpi	No induction of immune responses	Significant increase in *CXCLi2*, *IL-1β*, *IL-10*, *IFNγ*, and *IL-13* mRNA levels at 7 dpi; and *IL-1β* at 21 dpi
Systemic immunity		Significant increase in SAA concentration at 7 dpi Significant increase in specific IgY levels at 21 dpi	No significant in-crease in specific IgY levels at 21 dpi (compared to control)	Significant increase in specific IgY levels at 21 dpi

^a^ Cecal immunity was assessed only at 1 dpi and 7 dpi ; ^b^ hepatic immunity was assessed only at 7 dpi and 21 dpi.

## Data Availability

Data supporting reported results will be provided upon request.

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
