# Peer review of "Intestinal Colonization of Campylobacter jejuni and Its Hepatic Dissemination Are Associated with Local and Systemic Immune Responses in Broiler Chickens"

_microorganisms, 2023, doi:10.3390/microorganisms11071677_

Round 1
Reviewer 1 Report
1. Plz add novelty and added value to the introduction section
2. Plz add to the M&M section the composition of experimental diets and management
3. Conclusion please add a second conclusion and taken-home massage at the end of the abstract and discussion section
4 Plz add more references from 2023, if any
1. There are minor issues in EE, such as L 22, Serum as serum; thus Ms must be carefully checked
Author Response
We thank the Reviewers for the time taken to revise our article and for their positive comments that surely are improving the manuscript quality. We have addressed their comments and have made changes in the manuscript accordingly.
Reviewer 1
Comments and Suggestions for Authors
- Plz add novelty and added value to the introduction section
Reply: To emphasize the novelty of our work, we slightly changed the organization of the Introduction section to make the last paragraph clearer on the novelty of the article.
- Plz add to the M&M section the composition of experimental diets and management
Reply: This study is a follow-up to our previous study published in Frontiers in Microbiology (Chagneau et al., 2023), which reported diet formulation in supplementary data. Regarding management, we added some information in the article but we are stressed-out that details can be found in the previous study.
- Conclusion please add a second conclusion and taken-home massage at the end of the abstract and discussion section
Reply: As suggested, we added an additional conclusion at the end of the abstract and in the discussion section.
4 Plz add more references from 2023, if any
Reply: We added two references from 2023: Munoz et al. 2023, Poultry Science ; Genovese et al. 2023, Microorganisms.
Comments on the Quality of English Language
- There are minor issues in EE, such as L 22, Serum as serum; thus Ms must be carefully checked
Reply: We changed as suggested. Additionally, English revisions were already made prior to submission by a specialized firm
Reviewer 2 Report
Line 97: How many birds used in the experiment? Please give the information.
Pease give the imformation for growth performation.
Why cecal tonsils sample collected at 1 and 7 dpi, liver samples collected at 7 and 21 dpi, and the serum samples collected at 1, 7, and 21 dpi?
Line 99: What is the method of inoculated?Oral gavage?
After inoculated with Campylobacter jejuni, what the CFU number of Campylobacter jejuni in the cecum?
Why in the room 2 the number of Campylobacter jejuni D2008b is higher (107 vs 103)?
There are a lot of keywords.
If HE stained tissue (cecum and liver) slices can be provided, the results will be more intuitive.
Author Response
We thank the Reviewers for the time taken to revise our article and for their positive comments that surely are improving the manuscript quality. We have addressed their comments and have made changes in the manuscript accordingly.
Reviewer 2
Line 97: How many birds used in the experiment? Please give the information.
Reply: This study is a follow-up to our previous study published in Frontiers in Microbiology (Chagneau et al., 2023). Therefore, some details were omitted to avoid repetition. However, this information was added to make the current study easier to understand.
Pease give the information for growth performation.
Reply: As mentioned in our previous study, no significant difference in body weight was noted at 1, 7, and 21 dpi. We added this information to the manuscript.
Why cecal tonsils sample collected at 1 and 7 dpi, liver samples collected at 7 and 21 dpi, and the serum samples collected at 1, 7, and 21 dpi?
Reply: We collected cecal tonsils, liver samples and serum samples at 1, 7, and 21 dpi. However, we did not use all samples due to high costs. Therefore, we decided to evaluate gut immune responses at the onset of the colonization (i.e., 1 and 7 dpi) and hepatic immune responses when some livers were contaminated by C. jejuni (i.e., 7 and 21 dpi). This information was added to the manuscript.
Line 99: What is the method of inoculated?Oral gavage?
Reply: This information was added to the manuscript and can also be found in the previous study.
After inoculated with Campylobacter jejuni, what the CFU number of Campylobacter jejuni in the cecum?
Reply: These results appeared in our previous study (Chagneau et al. 2023).
Why in the room 2 the number of Campylobacter jejuni D2008b is higher (107 vs 103)?
Reply: We did a recap of the previously published study results at the start of the discussion. We also encouraged the reader a bit more to read the first study to get better insight of the experimental design.
There are a lot of keywords.
Reply: According to the author’s guidelines, we can list three to ten relevant keywords.
If HE stained tissue (cecum and liver) slices can be provided, the results will be more intuitive.
Reply: HE staining was not performed in this study. Tissues were used to perform RT-qPCR in order to assess changes in various chemokine/cytokine mRNA levels. We unfortunately do not have any tissue leftover to do it. It indeed is a good idea that we should implement in our next studies.
Reviewer 3 Report
The article being reviewed discusses an investigation of changes in the immune response of broiler chicken intestines upon inoculation by Campylobacter jejuni. The study looks at markers of immune protein in various tissues and organs through comparative analysis, and presents important scientific data that could be valuable to researchers and academics in this field.
However, for successful publication of this article, significant revisions should be made:
1. Presentation of results needs improvement; tables and figures lack variety and no one schema/table appears in the 'Results' section. To facilitate the perception of the received data it is necessary to add the table with the comparative analysis of changes for the different experimental groups of chickens.
2. Discussion of results is too detailed; consider adding schemas or illustrations depicting formation of immunity in chickens colonized by Campylobacter jejuni
3. An additional 'Conclusion' section must be added emphasizing significance of findings, practical applications (in production) and future research directions.
Author Response
We thank the Reviewers for the time taken to revise our article and for their positive comments that surely are improving the manuscript quality. We have addressed their comments and have made changes in the manuscript accordingly.
Reviewer 3
The article being reviewed discusses an investigation of changes in the immune response of broiler chicken intestines upon inoculation by Campylobacter jejuni. The study looks at markers of immune protein in various tissues and organs through comparative analysis, and presents important scientific data that could be valuable to researchers and academics in this field.
Reply: We wish to thank the Reviewer 3 for recognizing the importance of our work.
However, for successful publication of this article, significant revisions should be made:
- Presentation of results needs improvement; tables and figures lack variety and no one schema/table appears in the 'Results' section. To facilitate the perception of the received data it is necessary to add the table with the comparative analysis of changes for the different experimental groups of chickens.
Reply: To facilitate the understanding between our previous study and the current work, we decided to keep the same color for each experimental condition and figure. As suggested, we added two tables summarizing the main results of our previous study and the current study in order to facilitate the understanding and the perception of all these findings.
- Discussion of results is too detailed; consider adding schemas or illustrations depicting formation of immunity in chickens colonized by Campylobacter jejuni
Reply: We are the first to study the effects of intestinal co-colonisation by C. jejuni and its hepatic dissemination on local and systemic immune responses in broilers. Therefore, we feel that our results should be validated beforehand by other studies to create a schema that reflects reality. Also, as mentioned in the Introduction section, the local immunity seems to be dependent on the C. jejuni strain, the diet, the timing of exposure etc., complicating the task. But as more studies are reported, a review of the literature will be made possible and therefore a figure could be created. We think, given the actual state of research on immunity and Campylobacter, especially when considering hepatic spread, that such a figure would not, at this point, be accurate enough.
- An additional 'Conclusion' section must be added emphasizing significance of findings, practical applications (in production) and future research directions.
Reply: As suggested, we added an additional conclusion.
Round 2
Reviewer 2 Report
The author has answered all my comments and I have no further comments.
Reviewer 3 Report
I believe that the authors fully responded to my comments and carried all the necessary changes to the article. I suppose that this manuscript can be published.